# Analyses and Insights into Genetic Reassortment and Natural Selection as Key Drivers of *Piscine orthoreovirus* Evolution

**DOI:** 10.3390/v16040556

**Published:** 2024-04-02

**Authors:** Laura Solarte-Murillo, Humberto Reyes, Loreto Ojeda, Juan G. Cárcamo, Juan Pablo Pontigo, Carlos A. Loncoman

**Affiliations:** 1Laboratorio de Virología Molecular, Instituto de Bioquímica y Microbiología, Facultad de Ciencias, Universidad Austral de Chile, Valdivia 5090000, Chile; laura.solarte@alumnos.uach.cl; 2Departamento de Genética Molecular y Microbiología, Facultad de Ciencias Biológicas, Pontificia Universidad Católica de Chile, Santiago 8331150, Chile; reyes.zenteno@gmail.com; 3Laboratorio de Bioquímica Farmacológica, Virología y Biotecnología, Instituto de Bioquímica y Microbiología, Facultad de Ciencias, Universidad Austral de Chile, Valdivia 5090000, Chile; 4Interdisciplinary Center for Aquaculture Research, INCAR, Concepción 4030000, Chile; 5Laboratorio Institucional, Facultad de Ciencias de la Naturaleza, Escuela de Medicina Veterinaria, Universidad San Sebastián, Puerto Montt 5400000, Chile; juan.pontigo@uss.cl

**Keywords:** *Piscine orthoreovirus* (PRV), reassortment, segmented genome, virulence, genome diversity, positive selection

## Abstract

*Piscine orthoreovirus* (PRV) is a pathogen that causes heart and skeletal muscle inflammation in *Salmo salar* and has also been linked to circulatory disorders in other farmed salmonids, such as *Oncorhynchus kisutch* and *Oncorhynchus mykiss*. The virus has a segmented, double-stranded RNA genome, which makes it possible to undergo genetic reassortment and increase its genomic diversity through point mutations. In this study, genetic reassortment in PRV was assessed using the full genome sequences available in public databases. This study used full genome sequences that were concatenated and genome-wide reassortment events, and phylogenetic analyses were performed using the recombination/reassortment detection program version 5 (RDP5 V 5.5) software. Additionally, each segment was aligned codon by codon, and overall mean distance and selection was tested using the Molecular Evolutionary Genetics Analysis X software, version 10.2 (MEGA X version 10.2). The results showed that there were 17 significant reassortment events in 12 reassortant sequences, involving genome exchange between low and highly virulent genotypes. PRV sequences from different salmonid host species did not appear to limit the reassortment. This study found that PRV frequently undergoes reassortment events to increase the diversity of its segmented genome, leading to antigenic variation and increased virulence. This study also noted that to date, no reassortment events have been described between PRV-1 and PRV-3 genotypes. However, the number of complete genomic sequences within each genotype is uneven. This is important because PRV-3 induces cross-protection against PRV-1, making it a potential vaccine candidate.

## 1. Introduction

The salmon industry is one of the main sources of income in North Atlantic and South Pacific countries and has experienced unprecedented growth in Norway and Chile over the last three decades. It provides animal protein globally and contributes to local development [1]. However, the rapid expansion of the industry, coupled with increased stocking densities, has been shown to influence pathogen evolution [2]. Viruses are the second leading cause of fish mortality, and the emergence of highly virulent viruses in the salmon industry has increased over the past decade [3]. One of the most common viruses detected in both diseased and healthy farmed salmon is *Piscine orthoreovirus* (PRV), which is considered an emerging virus. PRV is the etiological agent of heart and skeletal muscle inflammation (HSMI) disease in *Salmo salar*, also known as Atlantic Salmon, as well as HSMI-like and circulatory disorders in other salmonid species such as *Oncorhynchus kisutch* and *Oncorhynchus mykiss*, also known as coho salmon and rainbow trout, respectively. The prevalence of PRV in farmed salmon during their time at sea exceeds 80% and mortality ranges from moderate to high, significantly affecting fish health and production [4,5].

First described in 2010 [6], PRV belongs to the family *Spinareoviridae*, genus *Orthoreovirus*, and infects fish only. PRV virions are non-enveloped and icosahedral, with a double-layered capsid approximately 70–80 nm in diameter [6,7]. The viral genome is 23 kilobases (kb) long and is located within the inner capsid. It contains 10 linear segments of double-stranded ribonucleic acid (dsRNA). The gene segments are divided into three sizes: long (L), medium (M), and small (S). The three long segments, L1, L2, and L3, encode the λ3 core RNA-dependent RNA polymerase, the λ2 core turret protein, and the λ1 core-shell protein, respectively. The three medium-length segments, M1, M2, and M3, encode the μ2 polymerase-associated protein, the μ1 outer capsid protein, and the μNS (non-structural) protein, respectively. Finally, as segment S1 is bicistronic, the four short segments S1, S2, S3, and S4 encode five proteins. The bicistronic segment S1 encodes the σ3 outer capsid and the p13 protein, whereas the monocistronic short segments S2, S3, and S4 encode the σ2 inner capsid, the σNS protein, and the σ1 attachment protein, respectively [7]. The S1 and M2 segments of the whole genome have been extensively used to characterize and classify PRV genotypes [8,9,10].

Phylogenetic analysis classifies PRV into three genotypes: PRV-1, including two variants, PRV-1a and PRV-1b [8,9,10]; PRV-2 [11]; and PRV-3, which includes two variants, PRV-3a and PRV-3b [12,13]. Depending on the viral genotype, host immune response, and environmental factors, PRV can cause various circulatory disorders in salmonids [14]. In *S.salar*, PRV-1 is the etiologic agent of HSMI disease, which is characterized by epicarditis, endocarditis, myocarditis, red skeletal muscle necrosis, and myositis [15]. HSMI was first described in Norway in 1999 and was linked to PRV a decade later [6]. Since then, PRV has been frequently detected in wild and farmed salmon in the Pacific Northwest and North Atlantic countries, as well as in farmed salmon in Chile [4]; however, the presence of viral RNA does not necessarily indicate disease development [16]. The PRV-3 genotype was described in 2013 as a new disease in *O. mykiss*, causing circulatory failure, ascites, anemia, and cardiac pathology known to be HSMI-like [17,18]. Currently, HSMI-like diseases have been described in farmed *O. mykiss* in Norway, Denmark, Scotland, the Czech Republic, and Chile [12,13,19,20] and have also been reported in farmed *O. kisutch* sampled from Chile [21]. In addition, PRV has been correlated with jaundice syndrome in *O.kisutch* and farmed Chinook salmon from Chile and British Columbia, respectively. However causality between jaundice syndrome and PRV has not been experimentally established [22,23,24]. The PRV-2 genotype has been detected and studied exclusively in *O. kisutch* in Japan and is associated with an anemic condition known as erythrocytic inclusion body syndrome (EIBS) [11]. Either way, the virus enters the bloodstream through the gastrointestinal tract and PRV targets red blood cells, where the replication cycle takes place [25,26].

The replication cycle of PRV begins with its entry into erythrocytes, facilitated by the interaction between the endosomal membrane and the outer viral capsid proteins, although the cell-specific receptors involved remain unknown [5]. In erythrocytes, viral replication peaks occur within the first four weeks of infection. Subsequently, the host anti-inflammatory response contributes to viral protein load reduction, tissue regeneration, and the alleviation of clinical signs (if present). At this stage, viral RNA remains in various circulating organs without significant replication and can take up to 14 months to clear depending on the host species [24,27,28]. In vivo experiments with PRV-1 have shown that the acute phase of replication of PRV-1a and PRV-1b variants is similar. However, certain PRV-1b variants, mostly from Norway, can cause severe cardiac lesions in *S. salar* (HSMI), in contrast to PRV-1a variants from Canada and the Faroe Islands [28,29]. This evidence suggests that PRV virulence varies as a result of selection and adaptation processes in intensive production environments that may drive viral evolution [2].

Segmented genome viruses, including PRV, possess a unique evolutionary mechanism that drives viral diversity through point mutations and reassortment, leading to antigenic drift and shifts, respectively [30]. Genetic reassortment can occur when a cell is either co-infected or superinfected with same-species viruses with segmented genomes, resulting in increased genetic diversity of viral progeny and potentially greater fitness for virulence relative to parental viruses. In the case of PRV, the presence of the virus in healthy salmon prior to the first reported outbreaks in 1999 was demonstrated by a retrospective analysis of Norwegian sequences of PRV-1 collected in 1988 and 1996. Molecular characterization of the S1 and M2 segments of these pre-outbreak PRV sequences clustered them within the low-virulence PRV variants from Canada [29]. In contrast, PRV sequences related to HSMI, such as the PRV-1b variant, had ten amino acid differences in the σ3 protein and seven amino acid differences in p13, both encoded by the S1 segment [29,31]. In the PRV-3 genotype, PRV-3a variant sequences detected in *O. kisutch* with jaundice syndrome in Chile were recently described to have unique polymorphisms not present in PRV-3b variant sequences from rainbow trout and coho salmon with HSMI-like diseases. These polymorphisms were found in proteins encoded by S1 and M2, with an amino acid difference in the σ3 protein showing positive selection (dN/dS > 1) [32]. However, the precise molecular mechanisms underlying the enhanced virulence of these PRV variants remain unclear, although it is hypothesized that reassortment and point mutation accumulation may contribute to the increased genetic diversity of viral progeny [31,32].

Access to large-scale genome sequencing has revolutionized comparative sequence analysis, allowing for the study of genetic reassortment, recombination events, and point mutation accumulation over time. The detection of genetic reassortments in viruses with segmented genomes often requires the identification of phylogenetic incongruencies [33]. Software tools such as the Recombination Detection Program (RDP) have been developed to detect reassortment events by identifying recombination breakpoints and similarities between aligned sequences using various statistical methods [34]. These methods have been successfully applied to detect genetic reassortments in other viruses belonging to the Sedoreoviridae family [35,36], as well as viruses with segmented genomes belonging to other families, such as the infectious pancreatic necrosis virus (IPNv) [37], infectious salmon anemia virus (ISAv) [38], and tilapia lake virus (TiLv) [39]. In this study, we aimed to analyze the potential for genetic reassortment and natural selection as key drivers of viral evolution, including the three PRV genotypes, corresponding to 28 concatenated genomes, using the most up-to-date collection of available complete genome sequences.

## 2. Methods

### 2.1. Collection of Sequences in Databases

The complete genome sequences of PRV available to date were downloaded from the GenBank database. Each segment was downloaded into separate files in the FASTA format and stored in individual files. The metadata associated with each genome are presented in Table 1 and Appendix A. The ten segments were individually aligned using the MUSCLE tool in MEGA X version 10.2 [40]. For recombination/reassortment analysis, the sequences of each complete genome were concatenated into a merged FASTA file by linking their respective FASTA headers. The sequences were organized according to the following segment lengths: L1 (4339 bp), L2 (3852 bp), L3 (4349 bp), M1 (2338 bp), M2 (2064 bp), M3 (2316 bp), S1 (1001 bp), S2 (1263 bp), S3 (1066 bp), and S4 (963 bp). The total size of the concatenated genome was 23,551 bp (Figure 1A).

### 2.2. Recombination/Reassortment Analysis

Reassortment events were identified using RDP5 software version 5.5 (RDP5 V5.5), which integrates multiple statistical methods based on phylogeny (BOOTSCAN, RDP, and SISCAN), substitutions (GENECONV, MAXCHI, CHIMAERA, and LARD), and distance comparison (PHYLPRO) to identify evidence of recombination [34]. A Bonferroni-corrected *p*-value of ≤0.05 was used. Recombination sites identified by four or more of the seven methods were considered significant recombination events, whereas events identified by three or fewer methods were considered putative recombination events. The start and end of the breakpoints identified by the RDP5 V5.5 were used to define putative recombinant sequences that were validated by phylogenetic analyses.

### 2.3. Overall Mean Distance and Selective Pressure Analysis

Non-coding regions were manually removed to align sequences from start to stop codon. Each segment was aligned against all other available sequences that belonged to a full genome. For the overall mean distance and selection Z-test, which uses the ratio of nonsynonymous to synonymous substitutions (dN/dS), each coding region was explored using the MEGA X version 10.2 [40] software with the default settings to compute the ratio using the Nei–Gojobori (Jukes Cantor) model/method and a bootstrap value of 1000 replicates. Additionally, to test the selection in each segment, the Hypothesis Testing Using Phylogenies (HyPhy) platform was used. For this, the codon-based alignments were curated and exported using Geneious prime 11 V 2023.2.1, Biomatters, New Zealand and further analyzed using Fixed Effects Likelihood (FEL) [41] and the Mixed Effects Model of Evolution (MEME) [42]. The parameters used to detect selection were set in the platform indicating to detect selection across sites, selecting either episodic (MEME was used) and pervasive selection (FEL was used). The default *p* value threshold was set to ≤0.1, and a more restrictive *p* value was manually set to ≤0.05.

**Table 1 viruses-16-00556-t001:** Metadata of the PRV complete genome sequences used in this study.

No.	Genotype/Variant	Sequence Name	Host	Year	Collection Country	Virulence	Observation	Reference
1	PRV-1a	NOR-1988	*S. salar*	1988	Norway	Healthy	Wild	[31]
2	PRV-1a	USMP_Coho_1993	*O. kisutch*	1993	USA	Healthy	Wild	[43]
3	PRV-1a	NOR-1996-V4105	*S. salar*	1996	Norway	Healthy	Wild	[29]
4	PRV-1b	NOR-1997	*S. salar*	1997	Norway	Disease (Unresolved)	Wild	[29]
5	PRV-1b	NOR-2005/TT	*S. salar*	2005	Norway	Disease	Farmed	[31]
6	PRV-1a	K31554	*S. salar*	2005	Canada	Healthy	Farmed	[43]
7	PRV-1b	PRV_050607.	*S. salar*	2007	Norway	Disease	Farmed	[44]
8	PRV-1b	CGA280-05	*S. salar*	2011	Chile	Disease	Farmed	[9]
9	PRV-1a	K31538	*Chinook*	2011	Canada	Healthy	Wild	[43]
10	PRV-1a	VT06062012-358	*S. salar*	2012	Norway	Disease	Farmed	[9]
11	PRV-1a	P.2-3_G460	*Chinook*	2013	Canada	Unknown	Farmed	[23]
12	PRV-1a	BCJ19943_13	*O. kisutch*	2013	Canada	Unknown	-	[45]
13	PRV-1b	NOR-2015/SSK	*S. salar*	2015	Norway	Disease	Farmed	[31]
14	PRV-1b	NOR-2015/MS	*S. salar*	2015	Norway	Disease	Farmed	[31]
15	PRV-1a	FO/41/16	*S. salar*	2016	Norway	Disease	Wild	[31]
16	PRV-1a	FO/1978/15	*S. salar*	2016	Denmark	Healthy	Farmed—smolt	[31]
17	PRV-1a	r17_631	*S. salar*	2017	Canada	Healthy	Farmed	[43]
18	PRV-1a	R2BC17	*S. salar*	2017	Canada	Unknown	Escaped—farmed	[43]
19	PRV-1b	NOR-2018/NL-V4105	*S. salar*	2018	Norway	Disease	Farmed	[29]
20	PRV-1b	NOR-2018/SF-V4105	*S. salar*	2018	Norway	Disease	Farmed	[29]
21	PRV-1b	mDa115	*S. salar*	2018	Norway	Disease	Farmed	[43]
22	PRV-1a	CAN-16-005ND-V4105	*S. salar*	2018	Canada	Healthy	-	[29]
23	PRV-3	DK/95-8109	*O. mykiss*	1995	Denmark	Unknown	Farmed	[13]
24	PRV-3	DK/PRV315	*O. mykiss*	2018	Denmark	Disease	Farmed	[13]
25	PRV-3	DK/PRV317	*O. mykiss*	2017	Denmark	Disease	Farmed	[13]
26	PRV-3	ADLPRV3	*O. kisutch*	2017	Chile	Disease	Farmed	[46]
27	PRV-3	NOR/060214	*O. mykiss*	2014	Norway	Disease	Farmed	[31]
28	PRV-2	PRV-2_Japan	*O. kisutch*	2012	Japan	Disease	Farmed	[11]

### 2.4. Phylogenetic Analysis

The sequences of the concatenated genomes, comprising segments L1, L2, L3, M1, M2, M3, S1, S2, S3, and S4, were aligned using the MUSCLE tool in MEGA X software version 10.2 [40]. Phylogenetic trees of each segment were constructed using the maximum likelihood (ML) method, along with the Kimura-2-parameter model and a bootstrap value of 1000 replicates. Phylogenetic trees and tanglegrams were generated using R Studio software version 4.2.2. The libraries to run the analyses were ape, phytools, dendextend, viridis, dplyr, and phylogram.

## 3. Results and Discussion

This study included twenty-eight concatenated genomes of PRV detected in wild and farmed salmonid hosts worldwide (Table 1). The concatenated genomes were trimmed to a length of 23,551 bp, as shown in Figure 1A. Of the total number of sequences analyzed, twenty-two concatenated sequences corresponded to the PRV-1 genotype, one to the PRV-2 genotype, and five to the PRV-3 genotype (Table 1). Based on this information, bioinformatic analyses identified 17 significant recombination/reassortment events, resulting in 12 reassortant variants: 11 belonging to the PRV-1 genotype (050607, CGA280-05, NOR-1997, r17631, NOR-2018/NL-V4105, NOR-1996-V4105, NOR-2015/MS, P.2-3 G460, NOR-2015/SSK, R2BC17, NOR-2015/MS) and one belonging to the PRV-2 genotype (Table 2). The five concatenated genomes of PRV-3 showed no recombination/reassortment events.

Significant reassortment events in PRV-1 were most frequently detected within the L1, L2, M1, M2, M3, S1, and S2 segments, with segment exchanges between the sequences of the PRV-1a and PRV-1b variants (Figure 1B). The L2, M2, S1, and S2 segments encode virion capsid proteins such as the λ2 core turret protein, μ1 outer capsid protein, σ3 outer capsid, and σ2 inner capsid, respectively [7]. The S1 bicistronic segment also encodes the cytotoxic integral membrane protein p13 [47]. The L1, M1, and M3 segments encode the λ3 RNA-dependent RNA polymerase, the μ2 polymerase-associated protein, and the non-structural protein µNS, respectively. RNA-dependent RNA polymerase λ3 is involved in viral genome replication, μ2 polymerase-associated protein is involved in transcription, and the non-structural protein µNS is involved in viral factory assembly and the early steps of viral replication [4,44]. Reassortments involving segments encoding both structural and nonstructural proteins contribute to an increase in the genetic diversity of viral progeny, allowing hybrid viruses to acquire antigenic variations that affect virulence, cell tropism, host range, and immune response evasion [33,48]. In vitro and in vivo studies with segmented genome viruses, such as the influenza A virus (IAv) and infectious pancreatic necrosis virus (IPNv), have shown that reassortment events are cell- and dose-dependent processes that occur in a non-random manner and can affect viral virulence [37,49]. Similarly, authors have found evidence of reassortment in isolates of tilapia lake virus (TiLv)—also a segmented genome virus that causes mortality rates of approximately 90% [50].

In this study, RDP5 V 5.5 software was used to detect and differentiate between recombination and reassortment events, as the latter involves the exchange of an entire segment (Figure 1). Nevertheless, in some cases, regions of the same segment appeared to recombine, as observed in the case of PRV-2-Japan and P.2-3 G460 sequences in the terminal region of the L1 segment, as well as R2BC17 and NOR-2015/MS sequences in the region of the L1 segment (Table 1 and Figure 1B). To confirm recombination events among the 28 PRV sequences, the same analysis was performed independently with the alignment of each gene segment. The results showed no significant recombination events for the nine segments of the PRV genome, except for the L1 segment. The L1 segment showed two significant recombination events in the NOR-2015/MS sequence, involving three regions of the L1 segment that are related to the NOR-1997 (minor parent) sequence (Appendix A).

The detection of phylogenetic incongruence has been linked to the detection of genetic reassortment. To detect an incongruence, a separate phylogenetic tree must be constructed for each viral RNA segment, and clades are identified based on monophyletic groups. If a viral variant has undergone reassortment, the phylogenetic trees will show the rearranged clades depending on the segment analyzed [33]. In this study, by constructing phylogenetic trees, we identified monophyletic clades grouping PRV-1, PRV-2, and PRV-3 genotypes when analyzing the S1–S4, M1–M3, and L2 segments (Figure 2 and Figure 3). Analysis of the whole genome sequence (concatenated) (Figure 4A), the L1 segment (Figure 4B), and the L3 segment (Figure 4C) did not cluster the PRV-2 genotype into a monophyletic clade like the other segments. Instead, the PRV-2 genotype clustered with the PRV-1a and PRV-1b sequences (Figure 4). These results suggest that if there were reassortment events in the evolutionary history of PRV, these events involved a gene exchange between unknown PRV-2 and PRV-1 parental sequences, as described by the RDP5 analysis (Figure 1). For PRV-3, all ten segments clustered the sequences in the same phylogenetic arrangement, indicating that the PRV-3 genotype has not yet shown evidence of genetic reassortment. However, it is important to note that in the present analyses, only five available complete genome sequences of the PRV-3 genotype were used to generate the tanglegrams and phylogenetic tree arrangements.

The S1 and M2 segments have been used to distinguish viral variants within the PRV-1 genotype (PRV-1a and -1b) (Figure 2) and also to determine differences in virulence between PRV-1a and PRV-1b in Canada, the Faroe Islands, Norway, and Chile [29,31]. Additionally, segments S4, L1, and L2 were incorporated alongside S1 and M2 to establish an alternative method for classifying PRV1 genotypes, affirming the presence of both high- and low-virulent isolates of PRV-1 in farmed Atlantic salmon in Norway [10]. In this study, the analysis of the other segments and the concatenated full genome revealed clade jumps between the PRV-1a and PRV-1b sequences (Figure 3 and Figure 4). For example, our results showed that the sequence NOR-2015/MS, classified as PRV-1b and detected in *Salmo salar* with multiple clinical signs of HSMI, as well as the sequence P.2-3 G460, belonging to the PRV-1a group and detected in healthy individuals, showed phylogenetic rearrangements in all ten segments analyzed (Figure 3 and Figure 4). Using the RDP5 software (version 5.5), the same sequences represent the minor parents from which the proposed reassorting regions may have originated (Figure 1B). Many of the sequences from fish with clinical signs of disease appeared to reassort some genomic segments with sequences from wild fish or fish without clinical signs (Table 2, Figure 1B).

Our results showed that reassortment events were mostly detected between sequences identified as different variants (PRV-1a and -1b), rather than different genotypes (PRV-1, PRV-2, and PRV-3). The absence of recombination and reassortment events between PRV variants could be explained by the geographic distribution of the virus-homotypic farmed hosts. Epidemiological studies that show cases of coinfection with PRV-1 and PRV-3 in farms from the same locality are scarce, reinforcing the hypothesis of rare interspecies transmission. However, in vivo experiments have shown that PRV-3 can also infect Atlantic salmon without causing the development of the HSMI disease, which was proposed as an alternative of cross-protection [51]. Curiously, in Chile, PRV-1 and PRV-3 was detected in coho salmon samples with clinical signs of jaundice syndrome suggesting natural co-infection in farming [22]. PRV-1 and PRV-3 had nucleotide similarities ranging from 76.5% to 80.9%, depending on the segment compared. Although these differences may explain the genetic incompatibilities that prevent the exchange of genetic material, more research and sequencing of PRV-3 are needed to draw broader conclusions in this regard. Nevertheless, reassortments should be monitored with caution as they usually lead to host-switching events. This has been demonstrated for other segmented genome viruses such as *Alphainfluenzavirus* and *Rotavirus A* [52,53].

The monitoring of naturally occurring genetic reassortment is complemented by in vitro assays, which have been a limitation for PRV due to the lack of a cell line capable of supporting successful PRV replication [54]. In 2015, an ex vivo culture platform for Atlantic salmon erythrocytes was demonstrated to enable the replication of a Norwegian PRV-1b isolate [55]. It is currently uncertain whether Canadian PRV-1a isolates exhibit the same replication kinetics in Atlantic salmon erythrocytes ex vivo. For future genetic reassortment studies, another alternative is to use a plasmid-based reverse genetics strategy to elucidate the mechanisms involved in reassortment, identify the segments most likely to be exchanged, and determine the replication kinetics of reassortants [56]. This approach could accelerate the development of new vaccines that combine segments from nonvirulent PRV-1a with virulent PRV-1b variants, similar to what has been achieved with live-attenuated rotavirus and influenza vaccines [56,57].

Point mutations could be followed by their effect on codons, meaning that they could be synonymous or non-synonymous. An indicator of positive selection is the detection of an increase in non-synonymous substitutions in codons that code for amino acids. We used two methods to explore the positive selection. First, we explored the positive selection over the entire sequence alignment. For this purpose, each codon was aligned from start to stop. Then, the overall mean distance and selection Z-test were used. Using this approach, we did not find any positive selections. However, to explore a more in-depth codon-based substitution analysis, we used Hypothesis Testing Using Phylogenies (HyPhy 2.5.33), an open-source software package to infer natural selection. In this study, we used the Fixed Effects Likelihood (FEL) method to infer non-synonymous (dN) and synonymous (dS) substitution rates on a per-site basis for a given coding alignment and its corresponding phylogeny. In particular, the FEL should be used when pervasive selection is suspected, as the selection pressure for each site is constant along the entire phylogeny [58]). Nonetheless, if episodic selection is suspected, then the mixed-effects model of evolution (MEME) is recommended, which employs a mixed-effects maximum likelihood approach to test the hypothesis that individual sites have been subject to episodic positive or diversifying selection. In our case, we performed both analyses and found that MEME detected more positive selection than FEL and that many of these codons are in structural proteins, such as the λ2 core turret protein (L2), the μ1 outer capsid protein (M2), and the σ3 outer capsid protein (S1). Interestingly, the μNS protein (M3) displayed high variability and positive selection (Table 3). Normally, structural proteins vary more than non-structural proteins, and the modification of structural proteins leads to changes in virulence. However, modifications in non-structural proteins also have a role over virulence [59].

The requirements for viral genetic reassortment include co-infection, genetic compatibility between recombinant molecules, and active replication [30,33]. Recently, PRV has been shown to induce persistent viral infections in erythroid progenitors of Atlantic salmon [16,27]. Experiments with PRV-persistent Atlantic salmon cell lysates have shown that the virus can infect naive fish. However, whether persistent PRV can cause reinfection and trigger new outbreaks in the field remains unknown [16,60]. If so, it is likely that the complete genomes of such persistent viruses provide ideal conditions for genetic reassortment upon PRV superinfection with different genotypes.

Some authors have pointed out that PRV has a reservoir in some wild salmonids and was naturally circulating in Atlantic salmon without causing outbreaks before farming began in Norway in the 1970s [61]. It is suggested that the possible presence of PRV-1 associated with HSMI must have occurred prior to 1999 [31]. The NOR-1997 sequence, obtained from a fish with an unresolved etiology, appears to be one of the relevant parental sequences in the process of understanding the evolution of PRV in the farmed environment. Moreover, a recent report indicated that PRV-1 has existed in Chile since 1994, manifesting as a genogroup of low virulence. Remarkably, this presence predates the initial HSMI outbreak in 2011 by 17 years [62]. In this context, PRV is considered a virus of interest for the study of viral evolution under selective pressure environments owing to host–virus–environment interactions. As a pathogen, PRV has characteristics such as an RNA genome with high mutation rates [31,32], the potential for genetic reassortment (as described in this article), and viral persistence. In addition, the fish environment includes high culture density and intensive fish farming; shortened rearing cycles; smoltification processes; saltwater acclimation; climate change with extremely low or high temperatures; and air, water, and ocean pollution. Together, these factors can drive the evolution of the viral genome and diseases caused by PRV, ultimately affecting the global salmon industry and the environment [2,61].

## 4. Conclusions

This study analyzed twenty-eight concatenated genomes of PRV from wild and farmed salmonid hosts worldwide. The results revealed significant reassortment events, particularly within the PRV-1 genotype, involving segments encoding virion capsid proteins, RNA-dependent RNA polymerase, and nonstructural proteins. These events contribute to the genetic diversity of PRV and have the potential to influence virulence, cell tropism, host range, and immune evasion. Notably, reassortment was not detected in the PRV-3 genotype, suggesting a different evolutionary trajectory for this genotype. These findings shed light on the underlying mechanisms driving PRV evolution and highlight the importance of monitoring reassortments in viruses with segmented genomes. Further research and surveillance efforts are needed to fully understand the mechanisms of genetic reassortment and its implications for PRV virulence, transmission dynamics, and the overall health of salmon populations. These findings will contribute to the development of effective disease prevention and control strategies in aquaculture.

## Figures and Tables

**Figure 1 viruses-16-00556-f001:**
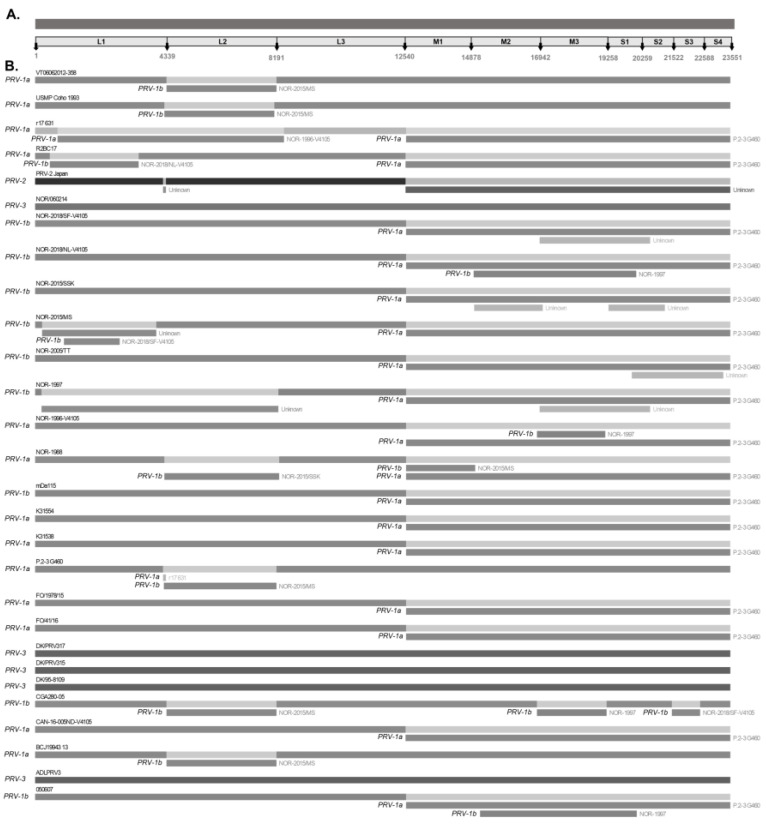
Recombination/reassortment events detected simultaneously by four or more methods using the RDP5 program. (**A**) The 23,551-base-concatenated PRV genome is used as an example of the nucleotide positions adjacent to each segment. Differences in shades of grey determine different viral variants to display mixtures (**B**) Recombination/reassortment events are shown along with the associated donor sequences. The segment belonging to each recombinant sequence is shown with its name as the “major parent”. Rectangles below the corresponding sequence represent the recombinant/reassortment segments associated with each event that originate from the “minor parent”. The “major parent” is usually a sequence that is closely related to the sequence that may have been the basis for most of the recombinant’s sequence. The “minor parent” is a sequence that is closely related to the sequences in the suggested recombinant region.

**Figure 2 viruses-16-00556-f002:**
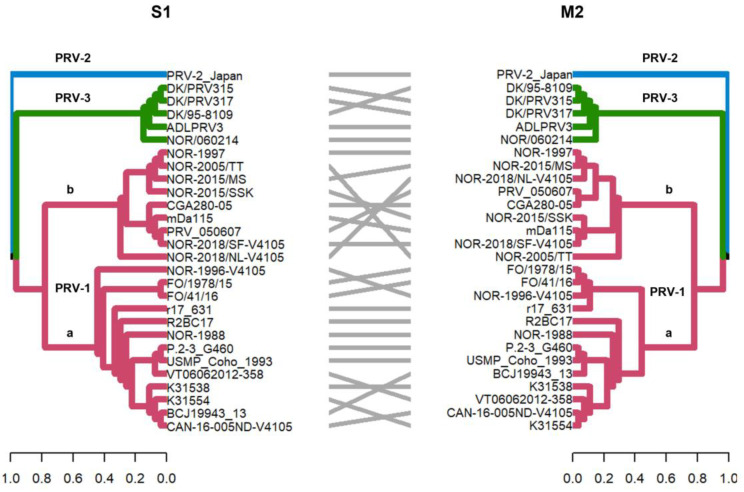
Phylogenetic analysis of segment S1 and M2 segments of 28 PRV sequences. Phylogenetic trees were constructed from nucleotide sequences, using the ML algorithm, bootstrap 1000, in the MEGA X program. The tanglegram shows the comparison of tree one (segment S1) with tree two (segment M2) where the PRV-1, PRV-2, and PRV-3 genotypes are grouped into monophyletic clades. In addition, these segments allow PRV-1 to be subdivided into the PRV-1a and PRV-1b viral variants, as previously described [12,29].

**Figure 3 viruses-16-00556-f003:**
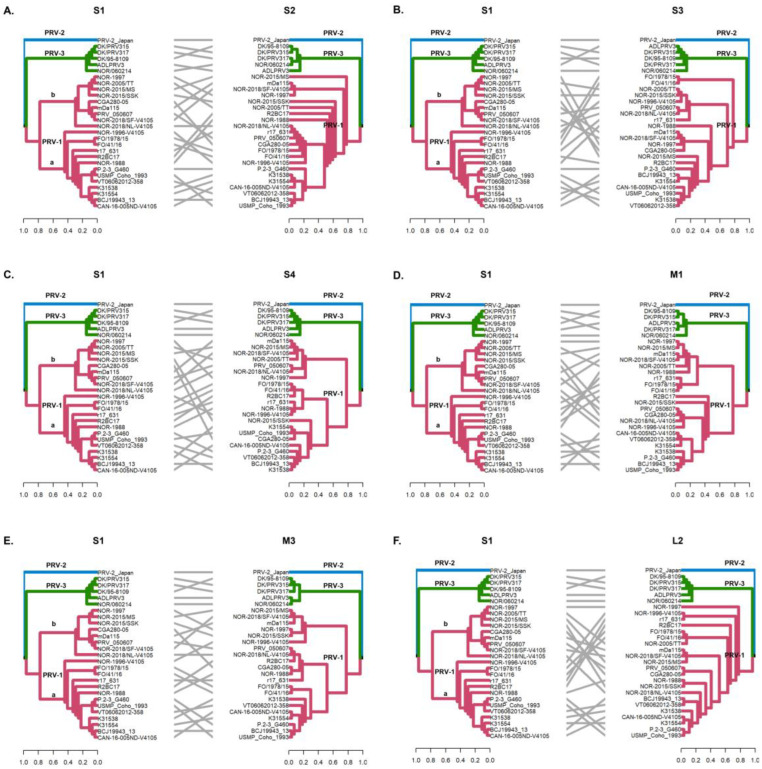
Phylogenetic incongruencies of 28 PRV sequences of segments S2 (**A**), S3 (**B**), S4 (**C**), M1 (**D**), M3 (**E**), and L2 (**F**) compared to segment S1. The sequence corresponding to PRV-2 and the sequences corresponding to PRV-3 remain grouped in their respective monophyletic clades, whereas the PRV-1a and PRV-1b sequences show clade jumping. Phylogenetic trees were constructed from nucleotide sequences, using the ML algorithm, bootstrap 1000, in the MEGA X program. The tanglegrams show, in all cases, the comparison of tree one (segment S1) with tree two: S2 (**A**), S3 (**B**), S4 (**C**), M1 (**D**), M3 (**E**), and L2 (**F**) segments, respectively.

**Figure 4 viruses-16-00556-f004:**
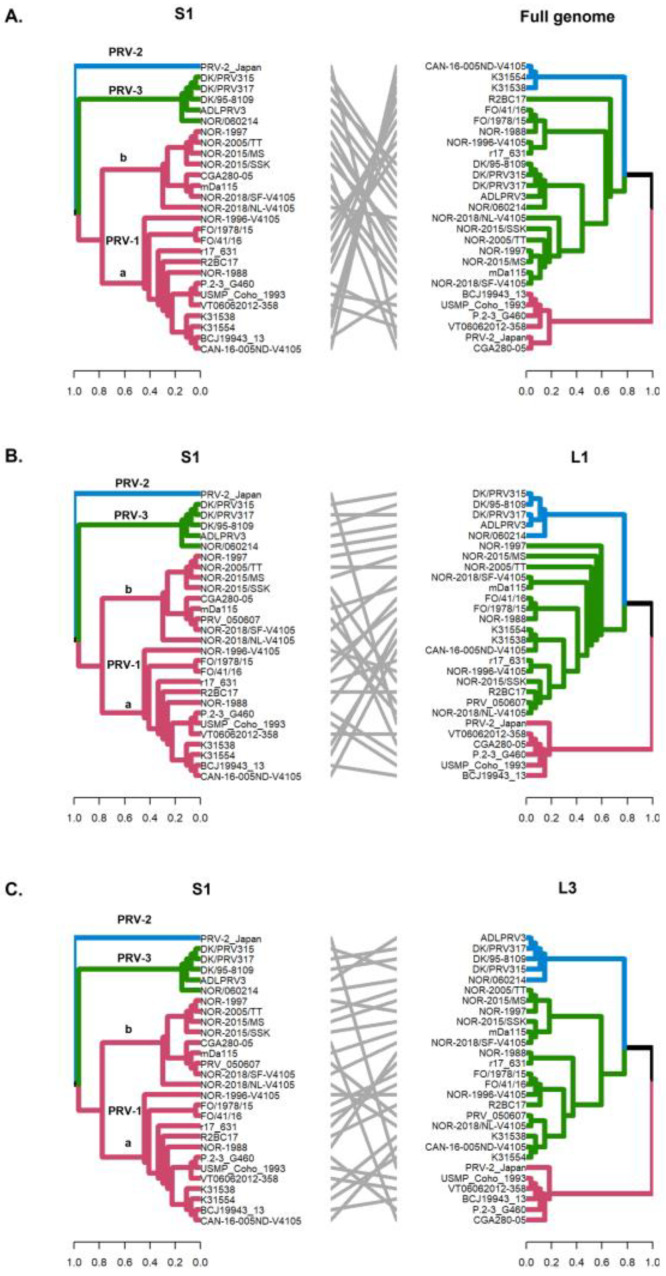
Phylogenetic incongruencies of the full genome (concatenated) (**A**), the L2 segment (**B**) and the M3 segment (**C**) of 28 PRV sequences compared to the S1 segment. The sequence corresponding to PRV-2, and the viral variants PRV-1a and PRV-1b are not grouped in monophyletic clades. Phylogenetic trees were constructed from the nucleotide sequences, using the ML algorithm, bootstrap 1000, in the MEGA X program. The tanglegrams illustrate in all cases the comparison of tree one (S1 segment) with tree two: full genome (**A**), L2 (**B**) or M3 (**C**) segments, respectively.

**Table 2 viruses-16-00556-t002:** Summary of recombination/reassortment events detected by the Recombination Detection Program 5 software version 5.5 (RDP5 V 5.5). Using a Bonferroni-corrected *p* cutoff value of ≤0.05, recombination sites identified by four or more of the seven algorithms in RDP were considered “significant recombination events” (*).

Event	Found in	Recombinant	Major Parent (% Similarity)	Minor Parent (% Similarity)	No. of Methods	*p*-Value Range	Position of Breaking Points (~)	Segments
1	17	50607 *	DK/PRV317 (79.4%)	P.2-3 G460 (98.4%)	7	6.6 × 10^−304^– 4.8 × 10^−20^	12,556–23,549	M1–M3 and S1–S4
2	5	CGA280-05 *	Unknown (NOR/060214)	NOR-2015/MS (99.3%)	7	1 × 10^−317^–5.3 × 10^−49^	4444–8182	L2
3	1	PRV-2 Japan *	USMP Coho 1993 (77.5%)	Unknown (ADLPRV3)	5	3.3 × 10^−174^–1.6 × 10^−30^	12,544–23,543	M1–M3 and S1–S4
4	1	NOR-1997 *	NOR-2015/SS (99.8%)	Unknown (NOR-1988)	7	1.9 × 10^−25^–3.2 × 10^−3^	223–8256	L1, L2
5	1	r17 631 *	NOR-1988 (99.9%)	NOR-1996-V4105 (100%)	5	2.6 × 10^−12^–4.3 × 10^−8^	754–8448	L1, L2
6	3	NOR-2018/NL-V4105 *	NOR-1996-V4105 (99.3%)	NOR-1997 (99.6%)	7	8.9 × 10^−14^–1.8 × 10^−4^	14,842–20,364	M2, M3, S1
7	1	NOR-1996-V4105 *	FO/41/16 (99.2%)	NOR-1997 (99.9%)	7	2.1 × 10^−10^–3.3 × 10^−3^	16,986–19,310	M3
8	1	NOR-2015/MS *	NOR-2015/SS (99.7%)	Unknown (NOR-1996-V4105)	4	1.0 × 10^−7^–2.3 × 10^−3^	227–4124	L1
9	1	NOR-1988	Unknown (NOR-1996-V4105)	FO/41/16 (99.6%)	3	9.2 × 10^−8^–1.7 × 10^−3^	754–8290	L1, L2
10	1	P.2-3 G460 *	PRV-2 Japan (77.6%)	r17 631 (100%)	7	9.2 × 10^−8^–6.2 × 10^−3^	4305–UND	Recombination in L1
11	1	CGA280-05 *	NOR-2018/NL-V4105 (56.8%)	NOR-2018/SF-V4105 (100%)	6	8.4 × 10^−7^–4.0 × 10^−2^	21,554–22,526	S3
12	1	PRV-2 Japan *	BCJ19943 13 (77.6%)	Unknown (NOR-1988)	7	4.9 × 10^−7^–1.9 × 10^−2^	4320–UND	Recombination in L1
13	3	NOR-1988	USMP Coho 1993 (56%)	NOR-1997 (99.1%)	3	3.2 × 10^−3^–1.5 × 10^−2^	17,190–19,422	M3
14	1	NOR-2015/SSK *	VT060620 (56.5%)	Unknown (r17 631)	4	4.0 × 10^−19^–7.5 × 10^−3^	14,872–17,189	M2, M3
15	1	NOR-1988 *	Unknown (K31554)	NOR-2015/MS (99.7%)	5	1.1 × 10^−16^–1.7 × 10^−3^	12,556–14,918	M1, M2
16	1	NOR-1988 *	r17 631 (99.9%)	NOR-2015/SSK (99.9%)	5	9.4 × 10^−10^–1.7 × 10^−2^	4375–8290	L2
17	4	NOR-1997	NOR-1996-V4105 (99.3%)	r17 631 (99.7%)	3	2.4 × 10^−4^–1.2 × 10^−2^	12,952–14,206	M1
18	1	NOR-2015/SSK	NOR-2018/SF-V4105 (99.8%)	050607 (99.6%)	2	8.7 × 10^−5^–5.8 × 10^−5^	658–7974	L1, L2
19	1	R2BC17 *	Unknown (FO/41/16)	NOR-2018/NL-V4105 (99.9%)	7	4.0 × 10^−10^–3.3 × 10^−3^	503–3532	Recombination in L1
20	4	NOR-2015/SSK *	P.2-3 G460 (56%)	Unknown (r17 631)	6	5.6 × 10^−9^–3.8 × 10^−3^	19,423–21,327	S1, S2
21	1	NOR-2005/TT	NOR-1997 (99.8%)	NOR-1996-V4105 (99.9%)	2	5.3 × 10^−3^–2.0 × 10^−2^	21,746–22,654	S3
22	2	r17 631	NOR-2018/NL-V4105 (98.8%)	Unknown (FO/41/16)	2	7.5 × 10^−2^–2.8 × 10^−2^	3602–8448	L1, L2
23	1	NOR-2015/MS *	Unknown (NOR-1996-V4105)	NOR-2018/SF-V4105 (99.9%)	5	3.6 × 10^−3^–2.0 × 10^−2^	986–2875	Recombination in L1

**Table 3 viruses-16-00556-t003:** Summary of positive selection pressure analysis using Hypothesis Testing Using Phylogenies (HyPhy 2.5.33) Fixed Effects likelihood (FEL) and Mixed Effects Model of Evolution (MEME). *p* value threshold default (≤0.1) and more restrictive (≤0.05) were used. Column 3 represents codons displaying positive selection; restrictive results values are shown in bold.

Segment (Protein)	Number of Codons	FEL Results Codon (Default/Restrictive)	MEME Results Codon (Default/Restrictive)
L1 (λ3 core RNA-dependent RNA polymerase)	1265	None	**11 (0.02)**; **34 (0.02)**; **81 (0.05)**; 216 (0.08); 320 (0.07); 658 (0.07);
L2 (λ2 core turret protein)	1178	952 (0.08)	**982 (0.02)**; **984 (0.02)**; **29 (0.03)**; **438 (0.04)**; **1037 (0.05)**; 592 (0.07); 964 (0.08); 359 (0.09); 822 (0.09)
L3 (λ1 core-shell protein)	1255	None	**720 (0.01)**; **86 (0.02)**; **111 (0.02)**; **328 (0.02)**; **567 (0.02)**; **17 (0.03)**; **51 (0.05)**; 794 (0.06); 206 (0.07); 149 (0.09); 932 (0.09)
M1 (μ2 polymerase-associated protein)	736	None	185 (0.08); 701 (0.09)
M2 (μ1 outer capsid protein)	673	656 (0.08); 225 (0.09)	139 (0.01); 369 (0.03); 191 (0.07); 466 (0.09)
M3 (μNS protein)	704	**481 (0.05)**; 101 (0.06); 532 (0.08); 483 (0.08); 505 (0.09); 502 (0.09)	**142 (0.01)**; **319 (0.01)**; **505 (0.03)**; **532 (0.03)**; **172 (0.05)**; **172 (0.05)**; 351 (0.06); 481 (0.07); 101 (0.08); 584 (0.08); 622 (0.08); 462 (0.09)
S1 (σ3 outer capsid protein)	328	**67 (0.04)**; 136 (0.06)	**67 (0.02)**; **85 (0.04)**; **72 (0.05)**; **136 (0.05)**; 203 (0.06)
S2 (σ2 inner capsid protein)	371	None	**13 (0.05)**; **332 (0.05)**; 21 (0.07); 337 (0.07)
S3 (σNS protein)	335	None	**317 (0.03)**; **119 (0.05)**; 306 (0.08)
S4 (σ1 attachment outer capsid protein)	313	None	254 (0.06); 252 (0.07)

## Data Availability

The original contributions presented in the study are included in the article/Appendix A, further inquiries can be directed to the corresponding author.

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
