# Peer review of "Analyses and Insights into Genetic Reassortment and Natural Selection as Key Drivers of Piscine orthoreovirus Evolution"

_viruses, 2024, doi:10.3390/v16040556_

Round 1

Reviewer 1 Report

Comments and Suggestions for Authors

This is an in silico work that analyzed twenty-eight concatenated genomes of PRV for genetic reassortment and point mutations. The study found that PRV-1 frequently undergoes reassortment events, that are leading to genetic diversity, antigenic variation and increased virulence.

L17-18 Latin names should be in italics.

L68 The S1 and M2 segments ……….. used to characterize and classify PRV genotypes. In Vatne et al 2021 the segments S4, L1 and L2   were used in addition to S1 and M2 to obtain a different method to classify PRV1 genotypes.  It should be mentioned.

L142 infectious pancreatic necrosis virus is abbreviated IPNV with a capitol V. How to write abbreviations of virus names is regulated by ICTV. Dont need to explain this abbreviation again in L235

L143 same here. infectious salmon anemia virus is abbreviated (ISAV)

L234 same here influenza A virus is abbreviated (IAV)

L313. related to its ability of PRV-1 to persist in the host and facilitate PRV-3 infection, allowing for a synergistic superinfection event rather than a co-infection and then reassortment. Do have any examples or reference of synergistic superinfection between PRV1 and-3? If not this is purely speculative and should be removed

L324 and elsewhere ex vivo should be in itaics

L334 The statement that RNA genomes undergo point mutations at higher rates than DNA viruses  is nor entirely correct, the mutation rate  of viruses with ssDNA genomes is higher than for several RNA viruses.

Author Response

Reviewer’s comments 1

This is an in silico work that analyzed twenty-eight concatenated genomes of PRV for genetic reassortment and point mutations. The study found that PRV-1 frequently undergoes reassortment events, that are leading to genetic diversity, antigenic variation and increased virulence.

Comment: L17-18 Latin names should be in italics.

Response: We have corrected all these typos throughout the manuscript L15-16-17-43-45-46-50-69-75-77-79-80-94-112-288-311)

Comment: L68 The S1 and M2 segments ……….. used to characterize and classify PRV genotypes. In Vatne et al 2021 the segments S4, L1 and L2   were used in addition to S1 and M2 to obtain a different method to classify PRV1 genotypes.  It should be mentioned.

Response: This information has been added to the Results and Discussion section in L281 - 284. This has only been included in the discussion as this section focuses on PRV-1 as mentioned in the paper and the introduction includes information on PRV-1, 2 and 3.

Comment: L142 infectious pancreatic necrosis virus is abbreviated IPNV with a capitol V. How to write abbreviations of virus names is regulated by ICTV. Don’t need to explain this abbreviation again in L235; L143 same here. infectious salmon anemia virus is abbreviated (ISAV); L234 same here influenza A virus is abbreviated (IAV).

Response: We have changed these names accordingly (L129 – 130; L222 – 224)

Comment: L313. “related to its ability of PRV-1 to persist in the host and facilitate PRV-3 infection, allowing for a synergistic superinfection event rather than a co-infection and then reassortment.” Do have any examples or reference of synergistic superinfection between PRV1 and-3? If not this is purely speculative and should be removed

Response: We have made changes to the manuscript based on the reviewer's recommendation.

Comment: L324 and elsewhere ex vivo should be in italics

Response: We have changed the manuscript throughout L15-16-17-43-45-46-50-69-75-77-79-80-94-112-288-311)

Comment: L334 The statement that RNA genomes undergo point mutations at higher rates than DNA viruses  is nor entirely correct, the mutation rate  of viruses with ssDNA genomes is higher than for several RNA viruses.

Response: We have made changes to the manuscript based on the reviewer's comments

Reviewer 2 Report

Comments and Suggestions for Authors

Comments on the Quality of English Language

Author Response

Reviewer’s comments 2

Review of Solarte-Muriollo et al. 2024

This work uses the genomic data already available for a fish reovirus (PRV) to look for reassortments,

with the help of a software. According (only) to probabilities, some putative reassortments are found

for PRV1 and PRV2, but not for PRV3. The topics is importance since this group of viruses are

certainly evolving due to the intense farming. The existence of very virulent and poorly virulent

strains is still unclear and, therefore needs more data. The subject is therefore of interest and the

paper is well written despite a need for minor corrections.

Without experimental data such as analyses of progeny of mixed infections, I would have

appreciated more precautions in the conclusions about the reassortments and recombination events

with low probabilities, for instance for ‘event’ n° 23 in table2. RDP can often find suspicions of

recombinations, that are not always strongly supported. Globally, the software has difficulties with

closely related sequences. Therefore, there could be artefact for PRV1a and PRV1b, and that would

also explain why no reassortment were found with more different sequences (PRV1 and PRV3).

About this last absence of detection, an element to discuss would be the global separation between

the trout and salmon farms (there are exceptions of course), and thus a low frequency of mixing

PRV1 and PRV3 in a same fish.

I would recommend publication but with major corrections about the hypothesis levels of the

findings.

Response: We agree with the reviewer that this group of viruses is under high selective pressure due to intensive farming. Regarding the reviewer's point about the RDP software, and in particular its difficulties with closely related sequences, we agree that the results shown by this software alone are not fully representative, and that despite the improvements made by Martin et al. published in Virus Evolution in 2021, there are still features of the software that could be improved. Nevertheless, this software has been used for more than 20 years and serves as a complementary tool to detect recombination and reassortment. In this sense, it is important to highlight that these results were complementary to phylogenetic analyses performed independently on a large dataset to show phylogenetic incongruence, which is known by several authors in the field as a feature indicating evidence of reassortment.

Furthermore, we slightly disagree with the reviewer's statement: Low frequency of mixing of PRV-1 and PRV-3 in the same fish. In fact. There is strong evidence that Oncorrynchus kisutch, also known as coho salmon, is frequently infected with these two genotypes during the fattening stage. This has already been published by Cartagena et al, 2020 and is a strikingly common finding in salmon farms in southern Chile.

Minor corrections:

Comments: The introduction is rather long; it might be shortened by removing a couple of unnecessary sentences, for instance l95-99 and l115-117, that can be moved to the discussion

Response: We have shortened the introduction by removing ideas and including them in the results and discussion section as requested by the reviewer L297 – 305; L314 – 315)

Comment: Line18. Latin names in italics

Response: We have changed the names to italics when appropriate. (L15-16-17-43-45-46-50-69-75-77-79-80-94-112-288-311)

Comment: L27. Replace ‘codify’

Response: We have replaced the word (L15 – 30)

Comment: L.34-35. repeats of previous sentences.

Response: We have modified the abstract to include the reviewer´s suggestion (L15 – 30)

Comment: L57. kb, not Kb

Response: This was modified throughout the manuscript

Comment: L61. ‘mu’ can be replaced directly by; same for sigma

Response: We have included these special characters in the manuscript (L55 – L63; L342 – L343) and in Table 3.

Comment: L70. 1a and 1b are rather two groups of variants or subgenogroups

Response: We have included the name variant for PRV1a and PRV1b in the manuscript (L65-67)

Comment: L185 and 198. 10000 replicates is a lot. If not a mistake (fig2 and 3 indicate 1000), is it necessary? would 1000 give the same results?.

Response: We have modified this in the manuscript, including 1000 bootstrapping only (L172; L185; 258; 268; 277)

Comment: Table2. p-values should be presented more clearly (one decimal should be enough). 6,626E-304-4,813E-20 is not clear to me.

Response: This has been modified in table 2

Comment: Figure 3 and 4 : trees are really small and difficult to read. Can their size be increased ?

Response: These images are meant to show phylogenetic incongruencies rather than specifics details about the trees. Nevertheless, the figures are at 600dpi resolution and could easily be zoomed in to see the names.

Comment: Figure 5 (and L.381-383) should be removed. It is too general and may be used for any aquatic pathogen. The impact of some factors on the virulence, such as environmental pollution, is not demonstrated, at least for PRV.

Response: We have deleted this figure as recommended by the reviewer.

Round 2

Reviewer 2 Report

Comments and Suggestions for Authors

Change IPNv in IPNV, same for TiLv, (etc.)

Add space between S.salar, and others (O.kisutch)